# Enhancing Visual Exploration through Augmented Gaze: High Acceptance of Immersive Virtual Biking by Oldest Olds

**DOI:** 10.3390/ijerph20031671

**Published:** 2023-01-17

**Authors:** Claudio de’Sperati, Vittorio Dalmasso, Michela Moretti, Emil Rosenlund Høeg, Gabriel Baud-Bovy, Roberto Cozzi, Jacopo Ippolito

**Affiliations:** 1Laboratory of Action, Perception and Cognition, School of Psychology, Vita-Salute San Raffaele University, 20132 Milan, Italy; 2Multisensory Experience Laboratory, Department of Architecture, Design and Media Technology, Aalborg University, 2450 Copenhagen, Denmark; 3RSA San Giuseppe, Associazione Monte Tabor, 20132 Milan, Italy

**Keywords:** healthy aging, oldest old, assisted living facility, virtual reality, virtual biking, visual exploration, augmented gaze, motion sickness, technology acceptance

## Abstract

The diffusion of virtual reality applications dedicated to aging urges us to appraise its acceptance by target populations, especially the oldest olds. We investigated whether immersive virtual biking, and specifically a visuomotor manipulation aimed at improving visual exploration (augmented gaze), was well accepted by elders living in assisted residences. Twenty participants (mean age 89.8 years, five males) performed three 9 min virtual biking sessions pedalling on a cycle ergometer while wearing a Head-Mounted Display which immersed them inside a 360-degree pre-recorded biking video. In the second and third sessions, the relationship between horizontal head rotation and contingent visual shift was experimentally manipulated (augmented gaze), the visual shift being twice (gain = 2.0) or thrice (gain = 3.0) the amount of head rotation. User experience, motion sickness and visual exploration were measured. We found (i) very high user experience ratings, regardless of the gain; (ii) no effect of gain on motion sickness; and (iii) increased visual exploration (slope = +46%) and decreased head rotation (slope = −18%) with augmented gaze. The improvement in visual exploration capacity, coupled with the lack of intolerance signs, suggests that augmented gaze can be a valuable tool to improve the “visual usability” of certain virtual reality applications for elders, including the oldest olds.

## 1. Introduction

Virtual reality (VR) is increasingly gaining traction as a rehabilitation tool for elders [1,2,3,4,5,6,7,8,9]. Indeed, the immersive character of VR yields the sensation of being inside an environment that can be navigated/explored through one’s movements (a sense of presence) and is an added value of the system. In producing a cogent realistic experience, it permits us to (re-)create situations otherwise unachievable for many elders. An example is virtual biking, an exertion game (exergame) that combines physical activity with a digital multisensory experience, where users pedal on a cycle ergometer while wearing a Head-Mounted Display (HMD) showing a 360-degree environment of a bike ride. Virtual biking can significantly contribute to elders’ healthy aging [10,11,12,13,14]. Although immersive visual exploration is a key ingredient of VR, many elders may not be able, or willing, to easily and frequently turn their head around because of possible neck or torso motor difficulties [15]. Indeed, neck pain has the highest prevalence rate among those aged between 60 and 85 years (higher in females), and is considered a serious public health problem [16,17]. Moreover, neck-related postural problems may be exacerbated by wearing an HMD [18]. Therefore, a system that enables users to immersively explore large portions of the visual field with small head turns in a semi-natural way (augmented gaze, i.e., amplified visual rotation contingent to head rotation) should facilitate active visual exploration, reducing the burden on the neck motor apparatus [19]. As in the case of other sensorimotor manipulations in virtual environments [20], augmented gaze can be achieved in an immersive system by increasing the visuomotor gain, i.e., the inverse ratio between head rotation and the ensuing visual rotation (Figure 1). Eye movements were not considered in this study; thus, the term augmented gaze refers to the visual amplification of head movements only.

The potential benefit of augmented gaze, i.e., increased “visual usability” due to amplified visual shifts and/or reduced head rotation, should be evaluated against possible negative side effects, as the visuomotor mismatch deriving from the amplification of the visual reafference of head rotation may not be well accepted. For example, prism adaptation, a visuomotor modification made popular by the seminal studies of Stratton [21,22], has a slower time-course in elders as compared to younger adults, and determines a longer after-effect [23]. Similarly, visuomotor adaptation, but not action sequence learning, is affected by age, suggesting cerebellar involvement [24,25]. Moreover, motion-related sensory conflict may induce motion sickness symptoms, especially in females [26], although the question of whether aging worsens motion sickness is still debated [27]. Thus, it is important to know whether augmented gaze can be utilized to increase the field of regard of older adults without discomfort [28]. This study addresses this issue.

## 2. Materials and Methods

### 2.1. Participants

Twenty participants (mean age 89.8 years, five males) participated in the study, 19 of them older than 80 and 15 of them older than 85. They were recruited through convenience sampling inside an assisted living facility (“RSA San Giuseppe”, Milan, Italy), where the study was conducted. None of them were self-sufficient, and none of them suffered from neck stiffness. In keeping with the principle that technology should serve as many individuals as possible, the only exclusion criterion was a cardiovascular risk. Table 1 shows the participants’ characteristics. Mental and motor conditions were assessed through the Mini-Mental State Examination (MMSE) and the Tinetti test, respectively, which are commonly used tools in assisted living facilities. For each participant, we used the last available measure, which in any case was taken not earlier than one month from the beginning of the study. Stay time (i.e., time since admission to the facility) was also registered. The study was approved by the Milan University ethical committee, and signed informed consent was obtained from all participants.

### 2.2. Procedure

Participants were seated on a chair or a (stabilized) wheel-chair, and had to pedal on a cycle ergometer while wearing an HMD (HTC Vive, HTC Corporation, New Taipei City, Taiwan, 110 degrees horizontal visual field) displaying a pre-recorded biking clip (360-degree video) under immersive conditions (Figure 2). The experimenters, with the help of facility personnel, helped them to feel as comfortable as possible with the experimental setup, especially when wearing the HMD, where care was taken to ensure good stimulus visibility and no discomfort. The cycle ergometer was set to zero resistance force to favour pedaling (only friction and inertial force were present). The video clip speed was not synchronized with the cycling cadence. A short familiarization session with gain = 1.0 (see below) was administered before the beginning of the experimental phase. Two biking video clips were used, both showing a bike ride through Copenhagen, and were randomly alternated across sessions. Footage (3D, 4K@30fps) was taken with a 360-degree camera (Insta360 Pro 2, Arashi Vision Inc., Shenzhen, Guangdong, China) mounted on the frontal seat of a trishaw bike. Participants were administered three sessions of virtual biking on three consecutive days, each session with an upper duration limit of 9 min. Each session had a different visuomotor gain (henceforth, gain): 1.0, 2.0, or 3.0. The former represents the natural gain (visual rotation mapped 1:1 to head rotation), whereas the other two gains implemented augmented gaze, obtained by linearly amplifying head rotation to yield 2-fold and 3-fold visual rotation, respectively. In the first session, the gain was always 1.0, while in the other two sessions, the gain was counterbalanced across participants. In this study, only the horizontal rotational component was amplified with augmented gaze (yaw, world coordinates). Participants were free to leave the session at any time for any reason. Data acquisition was controlled by a custom MATLAB script, which in turn launched a custom core executable code for visual presentation built under Unity3D.

### 2.3. Session Duration and Pedaling Duration

We measured both the session duration, which indicated whether participants opted to leave the session beforehand, and the actual pedaling duration, which indicated whether participants took pauses during the session. The latter was monitored by the experimenter using a chronometer; thus, it should be considered an approximate measure. The session duration distributions were negatively skewed, and significantly departed from normality at all gain values, as assessed with the Shapiro–Wilk test (always *p* < 0.01). Therefore, when using session duration as the dependent variable, the values were reflected and log-transformed before performing statistical analyses.

### 2.4. Cybersickness Symptoms

Just before the beginning and just after the end of each session, participants were administered the Simulator Sickness Questionnaire (SSQ, [29], a tool which is extensively used to measure cybersickness symptoms, particularly with elder populations [27], and for which a simple score interpretation is available [30]. The SSQ is composed of 16 items targeting three symptom clusters (nausea, oculomotor disturbance, and disorientation). We computed the differential SSQ score as the difference between the total SSQ score after and before each session as an indicator of the cybersickness symptoms resulting from virtual biking. Positive values indicate that symptoms increased after the virtual biking session. The differential SSQ score distributions at each gain value did not depart significantly from normality, as assessed with the Shapiro–Wilk test, except at gain = 1.0, where the distribution was positively skewed (*p* < 0.01). However, we obtained the same results by using the log-transformed scores ).

### 2.5. User Experience Ratings

After the responses to the post-session SSQ were collected, participants were also asked to rate satisfaction, motivation, sense of presence, and perceived safety. All responses were given on a 5-point numerical rating scale, and the questions were formulated as follows: (1) “How much did you like this experience?”; (2) “How much would you repeat this experience?”; (3) “How much did you feel like being biking outdoors?”; (4) “How much did you feel safe during virtual biking?”, where 1 represented no sensation at all and 5 represented full sensation, the other ratings indicating intermediate levels. For each participant, the four ratings were then averaged into a global user experience index. The global index distributions at each gain value did not depart significantly from normality, as assessed with the Shapiro–Wilk test.

### 2.6. Head Movement

Head pose signals were acquired from the HTC sensors and downsampled to 90 Hz. No low-pass filtering was necessary. Quaternions were transformed into instantaneous yaw, pitch, and roll angles (world-centered head co-ordinates). For each trial, we also computed the instantaneous visual stimulus yaw, pitch, and roll angles, obtained by multiplying the corresponding head co-ordinates by the current gain (world-centered visual coordinates). With the unitary gain, the two sets of co-ordinates coincide in Figure 3. A convenient indicator of explorative head movements was obtained by computing, for each trial, the mean absolute horizontal head angular deviation from the straight-ahead direction (mean yaw, world coordinates). Similar results were obtained by using the mean horizontal head rotation speed. The corresponding visual exploration indicator (the ensuing horizontal deviation of the visual field) was computed by applying the same procedure to the visual co-ordinates. Eye movements were not recorded. Because of their positively skewed distributions at all gain values, head rotation and visual rotation data were log-transformed before performing statistical analyses.

### 2.7. Data Analysis

All data were analysed through a mixed-models approach. To this end, we fitted the data to a linear model with gain, age, gender, MMSE score, Tinetti score, stay time, and session duration as fixed factors, and participants as random factors (intercept and slope). Almost identical results were obtained using pedaling duration as a predictor instead of session duration. When the dependent variable was either session duration or pedaling duration, session duration was removed from the fixed predictors. The variance inflation parameter (Vif) was always below 5, indicating that multicollinearity was not a relevant issue. The α level was set to 0.01. All analyses were performed with a custom MATLAB script. For the mixed-models analyses, we used the MATLAB fitlme function with the default options.

## 3. Results

### 3.1. Participants’ Acceptance

Participants appeared to be quite happy to perform the task, as inferred from informal conversations. This positive impression was confirmed by the satisfaction, motivation, presence, and safety ratings, which were on average very high (4.0, 3.8, 4.1, and 4.2, respectively), and by the SSQ differential scores, which were on average very low, with a mean value of 7.7, which is considered to indicate only “minimal” cybersickness symptoms (<10, [30]). Indeed, participants engaged quite effectively in virtual biking: the actual session duration was on average 24.7 min out of a maximal duration of 27 min of virtual biking (3 sessions of 9 min each), and only one participant opted to leave the study without performing the last pedaling session. Participants did not keep pedaling throughout the entire session duration, though, and paused frequently, often following the stops of the pre-recorded bike rides. On average, the pedaling time was 17.4 min. Participants also engaged spontaneously in active visual exploration, with a mean absolute horizontal head deviation from straight ahead of 5.5 degrees.

### 3.2. Effects of Gain

Crucially, gain did not significantly (always *p* > 0.5) affect participants’ global user experience index (Figure 4), although satisfaction seemed to decrease with increasing gain, though not significantly (*p* = 0.060). Similarly, SSQ scores (Figure 5) and session/pedaling duration (Figure 6) were not significantly affected by gain, although a non-significant slight tendency to decrease was apparent at increasing gains for the session and pedaling duration. Participants’ age generally did not significantly influence these measures, and neither did participants’ gender, MMSE score, Tinetti score, stay time in the facility, and session/pedaling duration, although there were a few exceptions. Appendix A report the results of the mixed-models analyses.

By contrast, gain significantly (always *p* < 0.001) affected both head rotation and the ensuing visual shift (Figure 7). The head rotation slope was −1.2 degrees for the gain unit, which means that, compared to the condition with gain = 1.0, horizontal head movements decreased on average by 18% for each gain unitary increase. Conversely, the visual rotation slope was 3.1 degrees for the gain unit, which means that, compared to the condition with gain = 1.0, horizontal visual exploration capacity increased on average by 46% for each gain unitary increase. For both head and visual rotation, the effect size was large (f${}^{2}$ > 0.35). We also computed the boundaries of visual exploration maximization and head rotation minimization (representing the extreme condition where visual exploration is maximized, with head rotation unchanged compared to unitary gain, and the opposite extreme condition where head rotation is minimized, with visual rotation unchanged compared to unitary gain, respectively). The upper boundary was obtained by multiplying the visual rotation values measured at gain = 1.0 by each gain value, and the lower boundary by multiplying the head rotation values measured at gain = 1.0 (identical to the visual rotation values at that gain) by each gain value. As can be seen from Figure 7, participants implemented a rather balanced strategy: at both gain = 2.0 and gain = 3.0, visual rotation and head rotation were about halfway between the reference line corresponding to gain = 1.0 (i.e., the no-change condition) and the respective boundaries (i.e., the maximal change conditions).

**Figure 5 ijerph-20-01671-f005:**
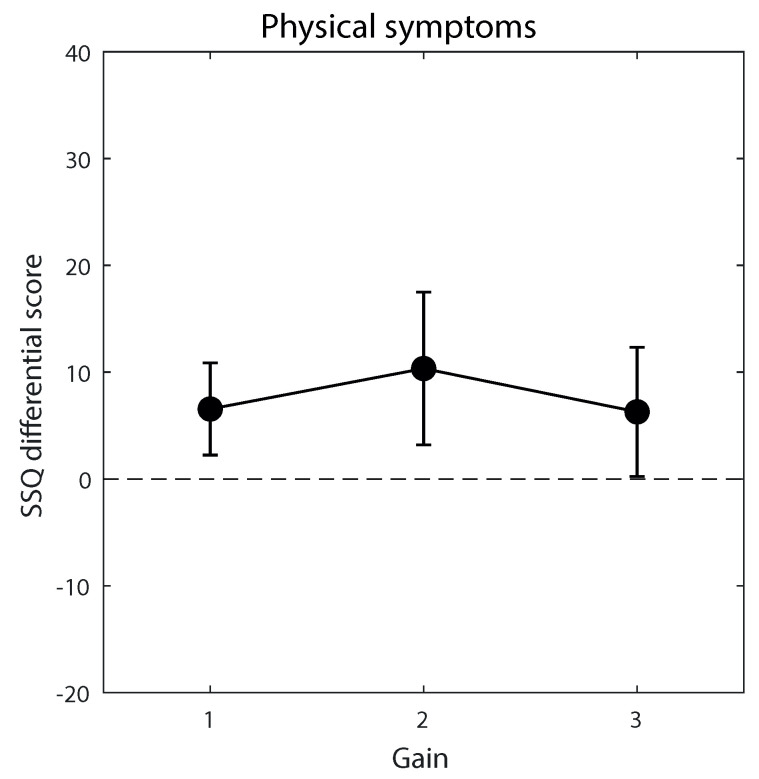
Physical cyber-sickness symptoms as assessed through the SSQ differential score (post-session—pre-session) across the three gain levels. The dashed line indicates a lack of symptoms resulting from the virtual biking session. The maximal score is 235.62. Error bars are 99% confidence intervals of the mean.

**Figure 6 ijerph-20-01671-f006:**
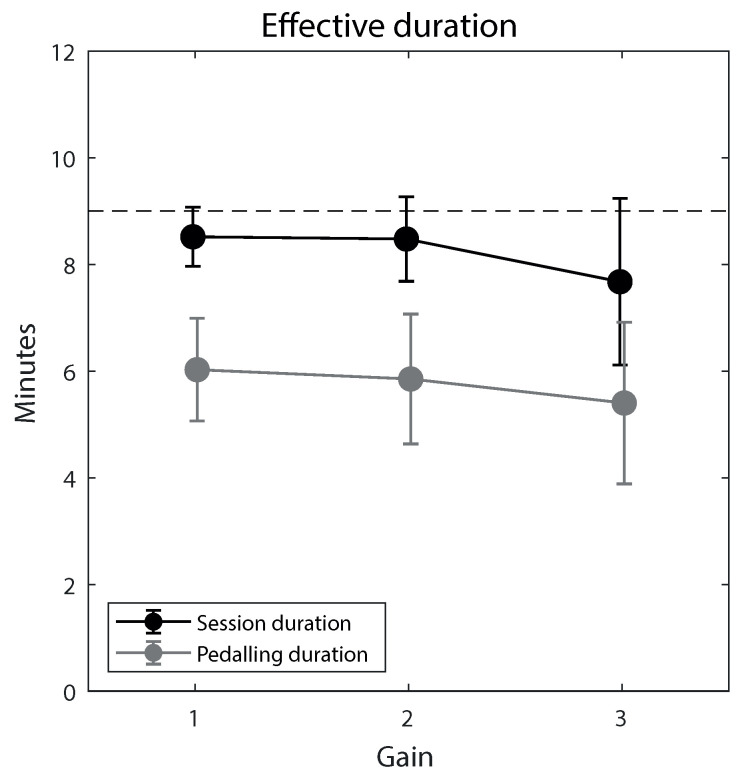
Effective duration of virtual biking sessions (black data points) across the three gain levels. Also shown is the effective pedalling duration (gray data points). The dashed line indicates the maximal session duration. Error bars are 99% confidence intervals of the mean.

**Figure 7 ijerph-20-01671-f007:**
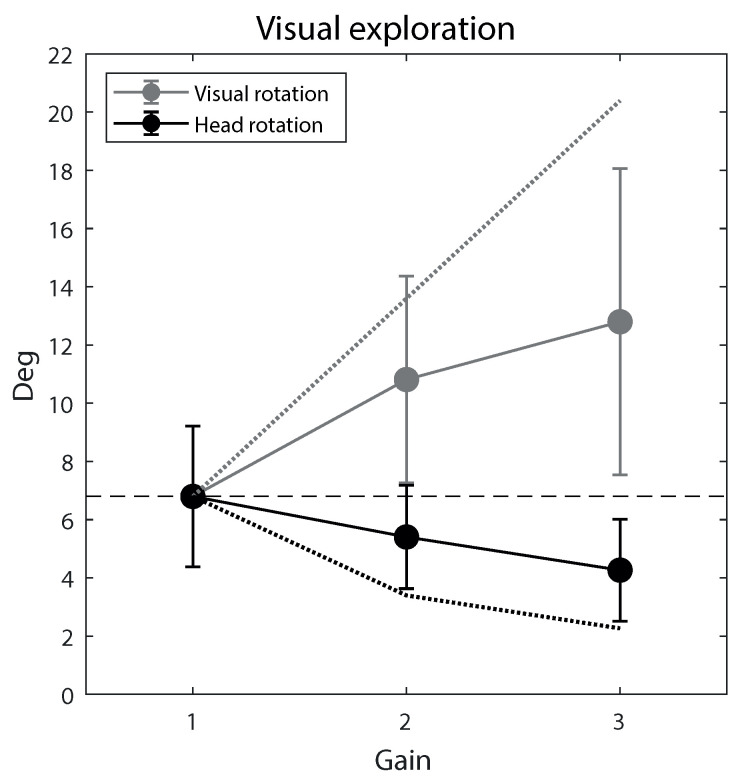
Visual exploratory activity across the three gain levels. Both the mean head rotation (absolute horizontal deviation from straight ahead, black data points) and the contingent mean visual rotation (grey data points) are illustrated. Error bars are 99% confidence intervals of the mean. The two dotted lines represent the condition in which visual exploration would be maximized, with no decrease in head rotation (grey), and the condition in which head rotation would be minimized, with no increase in visual exploration (black). Participants implemented a balanced, intermediate strategy. The dashed line indicates the natural condition (gain = 1.0), where the head and visual rotations coincide.

## 4. Discussion

The first consideration that emerges from this study is that the global user experience index, obtained by averaging satisfaction, motivation, sense of presence, and perceived safety ratings at the individual level, was on average very high, and cyber-sickness symptoms very low, thus pointing to high acceptance of this virtual biking experience by elders. The long session duration, a large proportion of which was spent pedaling, also agrees with this conclusion. The lack of effect of age on the considered dependent variables is similar: aging did not seem to weaken acceptance of virtual biking. This is important because, despite the increasing number of studies where VR is used with older adults, evidence about VR acceptance by the very aged is increasingly needed [31]. Indeed, the global population of the “oldest olds”—people aged 80 and older, according to the U.S. National Institutes of Health, or above 85, according to the British Geriatrics Society—is expected to more than triple between 2015 and 2050 [32]. In this regard, consider that the mean age of our participants was 89.8 years (range = 74–103), and that none of them were self-sufficient, which makes this study an important step towards understanding how VR technologies can be tailored to the oldest old population. The high acceptance of virtual biking by our participants may depend on the generic positive valence of introducing novelty into an otherwise routine life inside an assisted living facility, where novelties are rare. In this sense, every new opportunity would count as positive. However, another contribution to high acceptance could be the fact that this virtual biking experience resembles a telepresence experience [33], where users feel projected in the real world, not simply in a computer-generated world. As such, telepresence-like virtual biking may have an added value in that it combines environmental curiosity (e.g., not simply a new indoor exercise) with first-person engagement (e.g., not simply watching TV). This would be in line with current ideas on technology design for well-being—in particular, human–computer interaction—based on motivational theories [13,34].

A second consideration is that no clear signs of an impaired user experience or physical symptoms worsening emerged with augmented gaze (gain > 1). Note that the physical symptoms may even be somewhat overestimated due to demand characteristics (a sort of “nocebo” effect in SSQ [35]). This observation is important because it is not known how elders react to large visuomotor mismatch. Visuomotor adaptation is generally more difficult in elders [23,24,25], and motion-related sensory conflict is a possible cause of motion sickness [26]. Clearly, due to the relatively small sample size, we cannot rule out that a small effect of gain simply went undetected, perhaps visible as a slight decrease in motivation ratings and session/pedaling duration—but not in SSQ ratings—with increasing gain. At the very least, however, our data suggest that the above difficulties, if present, were of small magnitude and did not represent an important burden under augmented gaze conditions.

By contrast, the effect of gain on head rotation and visual rotation was a major effect, thus showing how elders reacted to augmented gain: they potentiated visual exploration capacity (by 46%) while concurrently reducing head motor effort (by 18%). Note that, in principle, participants might have implemented other strategies, such as maximizing visual exploration without reducing head movements, or minimizing head movements without increasing visual exploration (the two boundaries in Figure 7). The results indicate that participants implemented a balanced, intermediate strategy: visual exploration partially increased, and at the same time head rotation partially decreased, without clearly prioritizing either the visual or the motor advantage, as the mean values were half-way between no change and maximal change for both the visual and motor components of exploratory activity. Note that we tested only two gains and only linear amplification. It is entirely possible that higher gains and/or non-linear amplification (e.g., gain increasing proportionally to head deviation from the straight-ahead position) would yield different reactions. This remains to be tested.

Thus, the emerging picture is that the improvement in exploration efficacy triggered by augmented gaze was not met with significant negative side effects, if any. However, the improvement was not mirrored in the subjective ratings either (i.e., no more positive ratings with augmented gaze). All in all, our data show a neutral impact of augmented gaze on elders’ acceptance.

One possible interpretation is that a trade-off exists between positive and negative aspects which cancel each other out. For example, a positive appreciation of augmented gaze, perhaps associated with the need for optic flow sensory seeking [36], might compensate the discomfort of a potentially awkward experience, thus resulting in overall neutral ratings. Another interpretation is that elders are generally more insensitive to sensory stimuli [37], thus attenuating both positive and negative responses. The latter interpretation may be reflective of declining reactivity in elders, possibly accompanying social withdrawal and isolation. In the future, a comparison with a group of adult users could shed light on this issue.

Regardless of which interpretation does apply, we consider that the lack of clear signs of intolerance to augmented gaze is a positive outcome of the study, for it paves the way to experimenting with augmented gaze under more motivating conditions. On the contrary, very negative effects of augmented gaze, especially on SSQ, would advise against its use in future experiment. This does not exclude that signs of intolerance could manifest with long-term exposure to augmented gaze, a condition that we did not explore in this study.

A case where augmented gaze could be very useful and motivating is in improving the interactions between a bike rider and a biking buddy during group virtual biking or virtual tandem biking [13]. In this case, the motivation to maximize interpersonal relations may be an ideal motivation for elders to utilize augmented gaze, which would facilitate not only visual exploration of the surrounding landscape, as in the present study, but also eye contact with the biking buddy/buddies (social gaze, [38,39,40,41]) when the biking buddy/buddies is/are located behind or laterally relative to the bike rider. Under such potentially highly motivating conditions, it is possible that the neutral effect of augmented gaze on subjective feelings would then become positive. Explicitly targeting individuals suffering neck/head motor difficulties may also better show the benefit of augmented gaze as reflected in subjective ratings, an issue which we could not address in this study because the convenience sampling resulted in no participants with head/neck motor difficulties. Both aspects remain to be tested.

More generally, and pending further tolerance testing across different user categories and scenarios, we submit that augmented gaze could be a useful option that can be easily included in all HMD-based virtual reality systems, thus contributing to improve their usability and accessibility.

## Figures and Tables

**Figure 1 ijerph-20-01671-f001:**
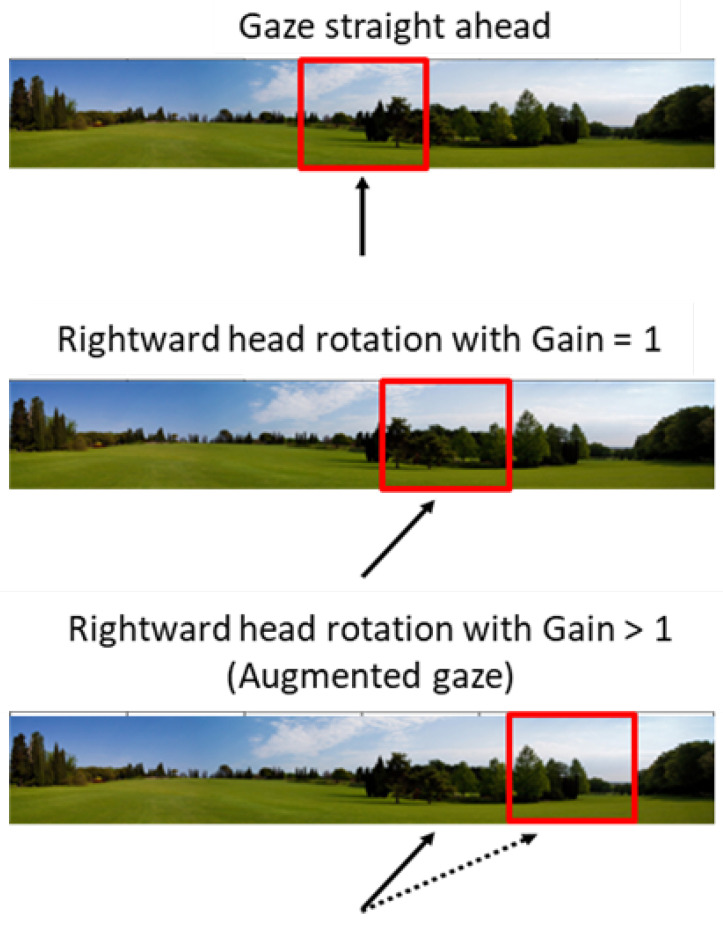
Augmented gaze. With the unitary gain, head movements and visual shifts are mapped 1:1 (i.e., the natural condition determined by head mechanics). When the visuomotor gain is higher, any given head movement produces a larger shift of the visual stimulus (visual amplification due to the augmented gaze condition). The red frame represents a hypothetical central region of the visual field, the solid arrow indicates the current head direction, and the dotted arrow represents the (larger) head movement that would have been required to produce that visual shift. Eye movements were not considered in this study; thus, the term augmented gaze refers to the visual amplification of head movements only.

**Figure 2 ijerph-20-01671-f002:**
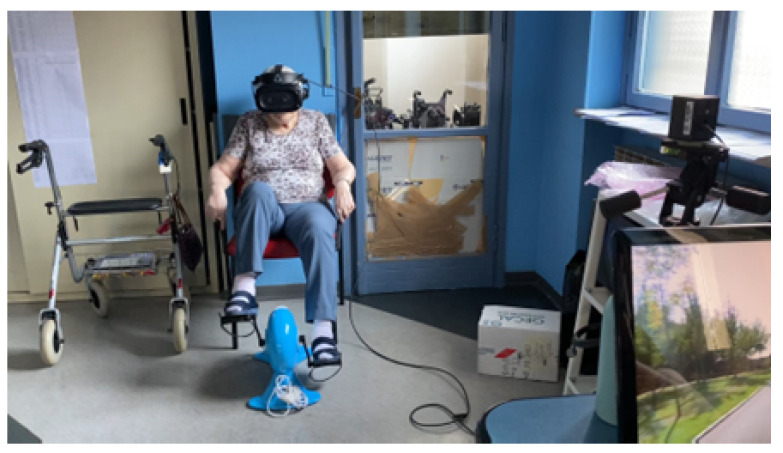
A photograph of a virtual biking session. Used with permission.

**Figure 3 ijerph-20-01671-f003:**
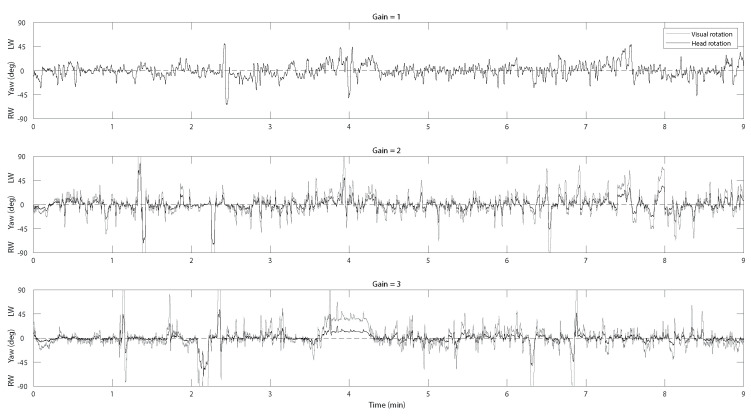
Example of head position recordings (black traces) during virtual biking in the 3 gain conditions in one participant (age = 81). The grey traces show the corresponding visual rotation (when the gain is 1.0, the two traces coincide). Only horizontal rotation (yaw) is illustrated. LW = leftward, RW = rightward. To facilitate comparison, the visual traces have been depicted with inverted signs (normally, visual stimulus rotation moves in the opposite direction to head rotation.

**Figure 4 ijerph-20-01671-f004:**
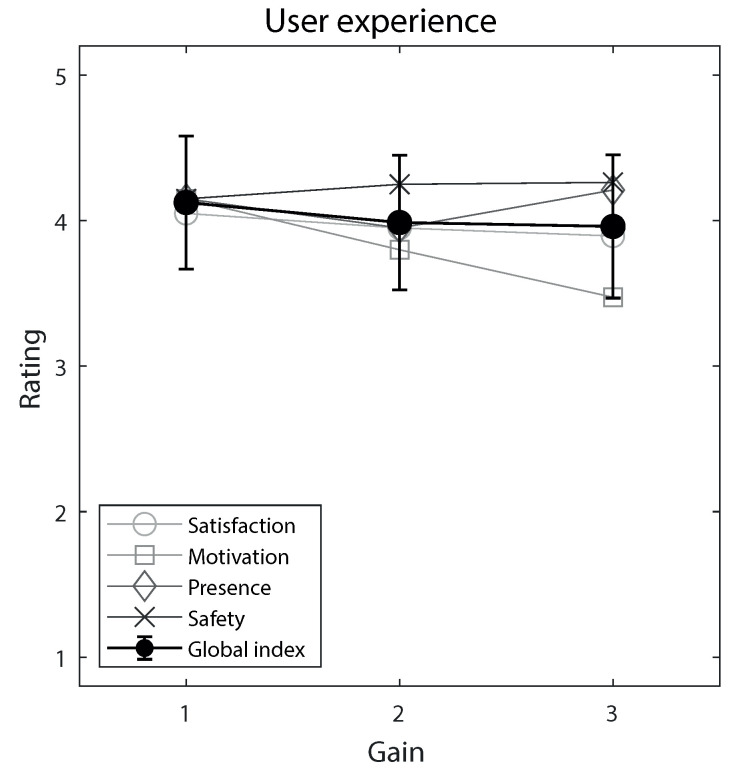
User experience, as assessed through the global index, across the three gain levels. The global index was computed by averaging, for each participant, the satisfaction, motivation, sense of presence, and sense of safety ratings (open symbols, range 1–5). Gain 1 is the natural visuomotor gain, whereas gains 2 and 3 implement two-fold and three-fold augmented gaze, respectively. Error bars are 99% confidence intervals of the mean.

**Table 1 ijerph-20-01671-t001:** Description of participant age, Mini-Mental State Examination (MMSE), and perception of balance and stability (TINETTI), shown as the mean (M) and standard deviation (SD).

Sample Characteristics	*M*	*SD*
AGE (years)	89.8	7.2
MMSE (score)	22.7	4.9
TINETTI (score)	17.1	5.5
STAY TIME (months)	20.9	16.6

## Data Availability

The data presented in this study are available upon reasonable request to the corresponding author.

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
