# Peer review of "Enhancing Visual Exploration through Augmented Gaze: High Acceptance of Immersive Virtual Biking by Oldest Olds"

_ijerph, 2023, doi:10.3390/ijerph20031671_

Round 1

Reviewer 1 Report

Summary

This paper focuses on the critical topic of the use of the immersive environment and XR technology to enhance visual exploration (augmented gaze) among older adults. Specifically, the participants participate in cycling on a cycle ergometer, where they wear a head-mounted device aimed to improve visual exploration. This was followed by two sessions where user experience, motion sickness, and visual exploration were measured. Findings suggest higher user experience, increased visual exploration, and decreased head rotation when using the head-mounted device. Although the study reports promising findings, I have a few minor concerns that I would like the authors to address to improve the quality of the manuscript.

Specific Comments

·         Throughout the manuscript, there are a few run-on sentences that I would suggest the authors split into two. For example, lines 19-22.

·         Line 68, should be “were seated…” or “were sitting…”

·         In figure 4, the lines are not discernable. I would recommend using solid, hashed, etc. I see one line decreasing linearly, but I am not able to understand which that one is. I would also recommend authors add that decreasing feature to discussions.

Author Response

Dear Reviewer 1

Thank you for your efforts in reviewing the manuscript. We have answered your concerns (marked in red) below:

  • Throughout the manuscript, there are a few run-on sentences that I would suggest the authors split into two. For example, lines 19-22.
    • Done. We have also made an additional review of the manuscript and identified other sentences. Thank you for the suggestion
  • Line 68, should be “were seated…” or “were sitting…”
    • Thank you. it is corrected 
  • In figure 4, the lines are not discernable. I would recommend using solid, hashed, etc. I see one line decreasing linearly, but I am not able to understand which that one is.
    • The figure was replaced. I would also recommend authors add that decreasing feature to discussions. The (non-significant) p-value was added for the apparent satisfaction decrease in the Results (lines 165-167) and commented in the Discussion (lines 225-230). Moreover, we have now supplied vectorized figures (eps-format) to increase readability. 

Best regards,

The authors

Reviewer 2 Report

This study is a good reminder of the fitness effect of sports based on virtual technology. From the perspective of commercial development, its research conclusions have strong social practice value, especially for middle-aged and elderly people. Home sports have become one of the necessary solutions for an aging society.

However, the elderly group was selected for the study, without highlighting the advantages and disadvantages of virtual technology for this special group. It is necessary to carry out comparative training experiments for different age groups to obtain more intuitive and effective scientific basis. In addition, as the direct causal relationship between visual effects and physical training needs to be further verified in medicine, it is expected that this research can conduct a more in-depth analysis in these aspects in the future.

Author Response

Dear Reviewer 2

Thank you for your efforts in reviewing the manuscript. We have answered your concerns (marked in red) below:

  • However, the elderly group was selected for the study, without highlighting the advantages and disadvantages of virtual technology for this special group. It is necessary to carry out comparative training experiments for different age groups to obtain more intuitive and effective scientific basis. In addition, as the direct causal relationship between visual effects and physical training needs to be further verified in medicine, it is expected that this research can conduct a more in-depth analysis in these aspects in the future.
    • We agree that more in-depth analysis is the logical next step, alas, not within this project's scope. We had already mentioned the need for future, more extensive comparisons (“pending further tolerance testing across different user categories and scenarios”, lines 280-281). Moreover, we have also specifically mentioned a comparison with an adult group (lines 257-258). We believe this to be sufficiently covered, since these hypothetical aspects would better be addressed in the light of empirical data, to be collected in future experiments.

Best regards,

The authors